# Long Interspersed Nuclear Element-1 Analytes in Extracellular Vesicles as Tools for Molecular Diagnostics of Non-Small Cell Lung Cancer

**DOI:** 10.3390/ijms25021169

**Published:** 2024-01-18

**Authors:** Emma C. Bowers, Alexandre M. Cavalcante, Kimberly Nguyen, Can Li, Yingshan Wang, Randa El-Zein, Shu-Hsia Chen, Min P. Kim, Brian S. McKay, Kenneth S. Ramos

**Affiliations:** 1Texas A&M Institute of Biosciences and Technology, Center for Genomic and Precision Medicine, Houston, TX 77030, USA; emma.ciel@gmail.com (E.C.B.); jessie.lican@gmail.com (C.L.); yingshanwang@tamu.edu (Y.W.); 2Department of Medicine, University of Arizona College of Medicine—Tucson, Tucson, AZ 85721, USA; cavalcante@email.arizona.edu; 3Houston Methodist Hospital Cancer Center and the Houston Methodist Academic Institute, Houston, TX 77030, USA; rel-zein2@houstonmethodist.org (R.E.-Z.); schen3@houstonmethodist.org (S.-H.C.);; 4Department of Ophthalmology, University of Arizona College of Medicine—Tucson, Tucson, AZ 85721, USA; bsmckay@eyes.arizona.edu

**Keywords:** LINE-1, non-small cell lung cancer (NSCLC), extracellular vesicles (EVs)

## Abstract

Aberrant expression of the oncogenic retrotransposon LINE-1 is a hallmark of various cancer types, including non-small cell lung cancers (NSCLCs). Here, we present proof-of-principle evidence that LINE-1 analytes in extracellular vesicles (EVs) serve as tools for molecular diagnostics of NSCLC, with LINE-1 status in tumor cells and tissues mirroring the LINE-1 mRNA and ORF1p cargos of EVs from lung cancer cell culture conditioned media or human plasma. The levels of LINE-1 analytes in plasma EVs from ostensibly healthy individuals were higher in females than males. While the profiles of LINE-1 mRNA and ORF1p in African Americans compared to Hispanics were not significantly different, African Americans showed slightly higher ORF1p content, and 2–3 times greater ranges of LINE-1 values compared to Hispanics. Whole plasma ORF1p levels correlated with EV ORF1p levels, indicating that most of the circulating LINE-1 protein is contained within EVs. EV LINE-1 mRNA levels were elevated in patients with advanced cancer stages and in select patients with squamous cell carcinoma and metastatic tumors compared to adenocarcinomas. The observed EV LINE-1 mRNA profiles paralleled the patterns of ORF1p expression in NSCLC tissue sections suggesting that LINE-1 analytes in plasma EVs may serve to monitor the activity of LINE-1 retroelements in lung cancer.

## 1. Background

A diploid human genome contains ~100 full-length copies of retrotransposition competent LINE-1 retroelements (Figure 1A) [1]. These retroelements are silenced epigenetically in nearly all healthy somatic cells by the interplay between DNA methylation and histone covalent modifications, a process orchestrated in part by retinoblastoma proteins and the NuRD corepressor complex [2]. Disruption of epigenetic silencing can unleash LINE-1 retrotransposition, which entails propagation of LINE-1 DNA, or other DNAs, through a copy-and-paste mechanism using an RNA intermediate. LINE-encoded proteins (ORF1p and ORF2p) exhibit *cis*- or *trans*-preference and bind mRNAs to form ribonucleoprotein particles (RNPs). These particles mainly localize to the cytoplasm or are stored in stress granules [3], but can also translocate to the nucleus where the endonuclease domain of ORF2p nicks a single strand of genomic DNA to expose a 3′-OH group that is used to prime and synthesize LINE-1 cDNA. This can lead to full-length or truncated insertions of LINE-1 or other sequences that modulate genome architecture and function.

The reactivation of LINE-1 retroelements can occur in several different contexts, particularly following DNA damage or disruption of epigenetic control by inflammation and oxidative injury [4,5]. Malignant transformation of somatic cells can also give rise to epigenetic disturbances that erode LINE-1 silencing and allow uncontrolled expression and accumulation of LINE-1 products [6,7]. Active LINEs are a source of endogenous mutagenesis, with reactivation in somatic cells causing a variety of genetic alterations, including aberrant splicing, exon skipping, gene fusions, and genome rearrangements that alter gene expression and cause genome instability [8,9,10]. Furthermore, LINE-1 reactivation creates a positive feedback loop that perpetuates the aberrant behavior of cells carrying mutations in tumor suppressor genes and/or oncogenes [11]. LINE-1 oncogenicity can also involve signaling pathways that are independent of retrotransposition [12]. 

Given the multifaceted roles of LINE-1 in cancer, we and others have postulated that readouts of LINE-1 activity may serve as indicators of oncogenic transformation. This hypothesis is supported by the strong correlation between LINE-1 expression and malignancy [6,13,14], tumor genomic instability [7,10], and cancer mortality [7,15]. LINE-1 DNA hypomethylation (i.e., activation) is also a common feature across many different cancer types [16], and correlates with poor clinical outcomes and lung cancer mortality [7,15]. The clinical utility of LINE-1 methylation status is limited because it depends on the direct testing of tissue biopsies using low-throughput technologies. 

Molecular biomarkers are sorely needed for lung cancer detection, especially because most cases are detected when curative interventions are no longer a viable option. Further, current screening with low-dose computerized tomography (LD-CT) is fraught with high false-positive rates leaving patients and providers with challenging follow-up decisions [17]. Non-small cell lung cancers (NSCLCs) are strongly impacted by LINE-1 dysregulation. LINE-1 reactivation is prevalent during early stage NSCLCs, especially in smokers [6,7,18], and the genome of NSCLC is strongly affected by LINE-1 insertions [13,19]. Thus, to harness the diagnostic potential of LINE-1 we have focused efforts on measurements of total plasma LINE-1 or LINE-1 analytes loaded onto extracellular vesicles (EVs) as lung cancer biomarkers. EVs are secreted membrane-bound vesicles containing curated DNA, RNA, and proteins from their cells of origin [20]. EVs are released by nearly all cell types and can be collected from saliva, blood, urine, and cerebrospinal fluid [21]. While the presence of LINE-1 products in EVs has been documented by us and others [22,23,24], detailed analyses to determine if LINE-1 products in EVs can be used for cancer detection and future development of point-of-care diagnostics have not been systematically completed.

With these goals in mind, we used a panel of transformed and non-transformed lung epithelial cell lines and human plasma to define the relationship between cellular and EV LINE-1 contents and to evaluate LINE-1 profiles in healthy and diseased states. Here, we report that the LINE-1 cargo in EVs isolated from conditioned media paralleled LINE-1 levels in the cells of origin, under both constitutive and carcinogen-inducible conditions. Among ostensibly healthy subjects, plasma EV LINE-1 content was higher in females than males and in African Americans compared to Hispanic Americans. Among subjects with NSCLC, considerable heterogeneity in plasma EV LINE-1 levels was observed when stratified by cancer stage, race, sex, and tumor type. Further, ORF1p levels in whole plasma using an ELISA platform closely approximated ORF1p and LINE-1 mRNA levels in Evs, indicating that the majority of circulating LINE-1 is contained in Evs. We conclude that measurements of LINE-1 analytes in Evs and plasma may be of value as liquid biopsies to monitor tissue level expression and activity of LINE-1 retroelements.

## 2. Results

To characterize EVs released by H520 lung cancer cells, EVs were isolated by PEG precipitation of conditioned media for 48 h. H520 cells have high constitutive expression of LINE-1 [24]. EV isolates ranged in diameter from 50 to 225 nm, with most particles concentrating at ~80 or 110 nm (Figure 1B), a range consistent with EVs of endocytic origin, mainly exosomes. The preparations were free of contamination with particles of a high diameter. EV protein profiles were examined by Western blotting, using unconditioned medium (UCM) and EV-free medium (EFM) as negative controls and total cell lysate as a reference control (Figure 1C). Traces of contaminating protein in EFM were removed by centrifugation coupled with several PEG resuspensions in fresh PBS as evidenced by the low abundance of ORF1p in EFM compared to the enriched EV sample. The exosome markers ALIX, Flotillin-1, and CD9 were enriched in EV fractions and absent in UCM and EFM. The absence of calnexin, a protein enriched in the endoplasmic reticulum, further validated the high quality of the EV preparations. 

Next, we examined the LINE-1 ORF1p content in EVs. ORF1p is a 40 kDa protein that preferentially forms multimers resistant to denaturation under reducing SDS-PAGE conditions (Figure 1A) [25,26]. The most abundant ORF1p species detected in H520 cell lysates were the monomeric and trimeric forms, while the dimeric form was predominant in EVs. These results are consistent with previous findings showing the ORF1p dimer in EV buoyant density gradients [24]. N-ethyl maleimide treatment did not enhance ORF1p detection under reducing conditions. Protein loading was verified by Ponceau staining (Figure 1D). Western blotting of EV isolates from H460 cells, another NSCLC cell line, challenged with the LINE-1 inducer benzo[a]pyrene (BaP), also yielded an ORF1p dimer in EVs (Figure 1E), confirming ORF1p EV export in two different NSCLC cell lines. 

To evaluate the presence of LINE-1 mRNA in EVs, intact H520 EVs were treated with RNAseA to remove contaminating RNA prior to lysis and isolation (Figure 1F). β-Actin mRNA was used as a positive control. RNA was also collected from UCM and EFM to monitor background levels. LINE-1 contains no introns; thus, LINE-1 primers cannot distinguish genomic DNA from cDNA. Thus, each LINE-1 sample was normalized to a matched control lacking reverse transcriptase (RTC) to account for residual gDNA signal. As the exon-junction spanning β-Actin primers do not amplify gDNA, the signal was normalized to a non-template control (NTC). The presence of both β-Actin and LINE-1 mRNA was confirmed in EVs, with significant elevations of LINE-1 (526 ± 67.2 FC) detected over the background and β-Actin (2887.6 ± 182 FC) (*p* < 0.05). LINE-1 and β-Actin mRNAs were not enriched in the EFM and UCM. 

### 2.1. Profiles of LINE-1 Abundance in NSCLC Cells and Their Corresponding EVs under Constitutive Conditions

We next determined whether EV LINE-1 content serves as a proxy of cellular LINE-1 levels (Figure 2). A panel of cells that included the non-transformed bronchial epithelial line, BEAS-2B, along with several NSCLC epithelial cell lines was examined (Figure 2A). Cells were allowed to condition EV-depleted media for 48 h before the collection of cell and media fractions. Western blotting showed constitutive ORF1p expression across all lines (Figure 2B). A549 cells exhibited the lowest levels of ORF1p, followed in ascending order by BEAS-2B, H441, H1299, H460, H827, and H520 cell lines. As A549 cells consistently exhibited the lowest LINE-1 levels, all subsequent LINE-1 measurements were expressed as a fold change relative to A549 cells or A549 EVs.

The next series of studies relied on PEG precipitation for EV isolation in order to facilitate comparisons between cells and plasma. PEG precipitation is incompatible with SDS-PAGE [27] (Appendix A), and therefore ELISA was used to measure ORF1p levels (Figure 2C, top row). ORF1p measures in cell lysates using the ELISA platform displayed rank order profiles comparable to those seen by Western blotting (Figure 2B). The BEAS-2B, H441, H1299, and H827 cell lines showed moderate ORF1p levels in EVs ranging from 2.6 to 3.2 FC relative to A549 EVs. The H460 and H520 cell lines had the highest ORF1p content with 6.7 and 7.8 FC, respectively. The linear regression showed a significant relationship between cellular and EV ORF1p content (Figure 2D, top, *p* = 0.03, *R*^2^ = 0.64), indicating that ORF1p content between cells and their corresponding EVs is proportional. 

Parallel analyses of LINE-1 mRNA expression in cells and Evs (Figure 2C, bottom row) showed A549 cells to have the lowest expression levels, followed by H460 cells 1.30 FC, BEAS-2B (1.35), H441 (2.84), H827 (2.92), H1299 (3.97), and H520 (9.57). In Evs, H460 cells exhibited 5.29-fold enrichment relative to A549 followed by BEAS-2B (8.44), H441 (12.79), H520 (28.26), H827 (32.44), and H1299 (56.89). Log2 transformation followed by non-linear Spearman’s regression (Figure 2D, bottom) showed a significant relationship (*r* = 0.89, *p* = 0.01) between these two variables. Curve fitting yielded a sigmoidal relationship (*r* = 0.90) which may be artifactually driven by H520 cells or, alternatively, implicate loading constraints of LINE-1 mRNA in cells with high LINE-1 content. Together, these results demonstrate that constitutive expression of LINE-1 in cells can be approximated by measurements of EV LINE-1 cargo. 

### 2.2. Cellular and EV LINE-1 Levels under Inducible Conditions

To determine if the remarkable cell-specific EV profiles seen under constitutive conditions were preserved upon the induction of LINE-1, cells were challenged with BaP, a known lung carcinogen and LINE-1 inducer. H460 cells were exposed to different BaP concentrations and allowed to condition their medium for 48 h (Figure 3). We have previously shown that BaP treatment does not change the number of secreted EVs, though a minor shift in the size of exosomes was noted [24]. There was not a statistically significant change in the number of BAPs secreted after BaP exposure, but it did appear that the size pattern of EVs was altered. See data below. 

Carcinogen treatment did not compromise cell viability (Figure 3A), but readily induced LINE-1 protein (Figure 3B). ORF1p was quantified in cells and EVs using the ELISA platform (Figure 3C, top row) and expressed as FC relative to DMSO. Cellular ORF1p exhibited concentration-dependent induction profiles (*p* < 0.05), with a 5.04 ± 0.16 mean fold induction at 1 µM BaP and a 4.92 ± 1.62 mean fold induction in EVs at the same concentration. ORF1p increases in BaP-treated cells were proportional to their corresponding EVs, as established by linear regression (Figure 3D, top; *p* = 0.03, *R*^2^= 0.95).

Cellular LINE-1 mRNA also increased as a function of increasing BaP concentrations (Figure 3C, bottom row), with 1.43, 1.80, and 1.95-fold increases seen in cells treated with 0.25, 0.5, and 1 mM BaP, respectively (*p* < 0.05). Mean LINE-1 mRNA in EVs exhibited a concentration-dependent trend, but the response was variable and not significantly different from DMSO. There was, however, a significant relationship between mean cellular and mean EV LINE-1 mRNA and ORF1p levels (Figure 3D, bottom; *p* = 0.048, *R*^2^ = 0.91). Together, these results demonstrate that fluctuations in cellular LINE-1 expression following carcinogen induction are mirrored by their corresponding EVs. 

### 2.3. LINE-1 in EV Isolates from Human Plasma of Ostensibly Healthy Individuals

We first examined the abundance of LINE-1 ORF1p in ostensibly healthy subjects to evaluate the degree of inter-individual variability (Figure 4). Twelve subjects (six African Americans and six Hispanics) were matched by age within 2 years, sex, and race (Figure 4D). EVs were isolated by ultracentrifugation and normalized by total plasma volume (3 mL). For each subject, half of the preparation was used for measurements of LINE-1 mRNA and for quantification of ORF1p. EV proteins were visualized by Western blotting (Figure 4A). The EV markers Annexin 2, Flotillin-1, and ALIX were used as positive controls. ORF1p dimer was present in all donors and exhibited significant variation across the cohort. Protein levels and a Ponceau stain are depicted in Appendix A. β-Actin, a positive control, was detected in all samples, while LINE-1 mRNA could only be detected in trace amounts relative to the RTC. Despite low detection in human EVs, a positive correlation between EV LINE-1 mRNA and EV ORF1p levels was found (Figure 4C; *p* = 0.004, *R*^2^ = 0.58). This relationship may reflect proportional EV cargo loading of ORF1p and LINE-1 mRNA, arguably in the form of high affinity LINE-1 ribonucleoprotein complexes [28,29,30,31]. 

In follow-up studies, we used densitometric measurements of ORF1p by Western blots normalized to total protein or mRNA levels to evaluate differences in LINE-1 EV cargo in our cohort by sex and race/ethnicity. ORF1p EV content was higher and exhibited a broader range in females than in males, with a mean ORF1p level of 94.01 arbitrary units versus 21.17 in males (*p* = 0.04). Mean LINE-1 mRNA levels were generally higher in females, with EV LINE-1 mRNA content in females at 0.26 log2 FC relative to RTC, which was borderline significant (*p* = 0.056) compared to males, with a mean of 0.053 log2 FC relative to RTC. While the profile of LINE-1 mRNA or ORF1p in African Americans compared to Hispanics was not significantly different, African Americans had slightly higher ORF1p content, and 2–3 times greater ranges of LINE-1 values compared to Hispanics. As we normalized EV inputs to total plasma volume, analyses were completed to rule out that the patterns observed were attributed to variations in EV abundance. Indeed, total protein levels exhibited no significant differences between groups (Appendix A). Together, these results reveal that EV LINE-1 content varies considerably between individuals. Importantly, these findings can inform the creation of larger range finding studies to evaluate EV LINE-1 patterns in cancer patients.

### 2.4. Relationship between Whole Plasma and EV ORF1p Levels

Our previous studies noted a relative absence of free LINE-1 products in plasma compared to the highly enriched quantities found in plasma EVs [24]. Thus, we hypothesized that the majority of circulating LINE-1 may be present in EVs, and that these levels may be approximated using whole-plasma ELISA measurements. To test this hypothesis, we measured bulk plasma ORF1p in ostensibly healthy subjects using ELISA and compared these values to measurements of LINE-1 mRNA (Figure 4G) and EV ORF1p (Figure 4H). Remarkably, a significant relationship between bulk plasma ELISA and EV LINE-1 mRNA levels was found (*R*^2^ = 0.41, *p* = 0.024), along with a near-significant relationship with EV ORF1p levels (*R*^2^ = 0.30, *p* = 0.063). These results support the hypothesis that circulating LINE-1 is predominantly contained within EVs and that performing a whole-plasma ELISA may approximate these values with a reasonable degree of fidelity.

### 2.5. EV LINE-1 mRNA in EVs Isolated from Lung Cancer Patients

Using a single-blinded approach, LINE-1 mRNA was measured in plasma EVs from 28 patients who underwent lung resection surgery for a lung lesion. EVs were isolated from 250 μL of plasma and processed for measurements of LINE-1 mRNA. The presence of EVs in these preparations was verified by NTA (Figure 5A). As a quality control, samples from six donors with less than 2300 particles and a β-Actin Ct greater than 38 were excluded from further analysis (Figure 5B). Stratification of EV mRNA profiles based on cancer stage and histopathology showed that Stage 1 subjects had lower EV LINE-1 levels than those with Stages II and IV (Figure 5C). Notably, not all Stage II and IV tumors exhibited high LINE-1 levels suggesting that the heterogeneity of response may be present. While mean LINE-1 levels did not significantly differ between squamous cell carcinomas and adenocarcinomas, a trend for higher levels in squamous cell carcinomas and metastatic tumors emerged (Figure 5D). The profile of EV LINE-1 mRNA content in NSCLCs paralleled the patterns of ORF1p expression in NSCLC tissue sections compared to non-tumor tissue (Figure 5E).

## 3. Discussion

Aberrant expression of the oncogenic retrotransposon LINE-1 has emerged as a hallmark of NSCLC. Most of the evidence to date has arisen from measurements of LINE-1 encoded ORF1p in cancer tissues at the time of resection, thus limiting the strength of inferences that can be made about the trajectory of illness, treatment response, or clinical outcome. To overcome these limitations, we have proposed measurements of LINE-1 cargo in human plasma-derived EVs as proxies of tissue LINE-1 expression. Here, we present proof of concept evidence that the LINE-1 cargo in EVs isolated from NSCLC cell lines mirrors their cellular content, and that measurements of LINE-1 analytes in plasma EVs can be used to stratify healthy subjects by gender and possibly race/ethnicity, and lung cancer patients by stage and histological type. 

EV cargo need not be consistently proportional between cells and EVs as the loading of DNA, RNA, and protein may be subject to regulation via multiple biochemical pathways, and influenced by active and passive export mechanisms [20,32]. Thus, we initiated systematic studies to evaluate the levels of LINE-1 analytes in EVs and their cells of origin. Our findings under both constitutive and inducible conditions revealed that the proportionality between cellular and EV cargo is maintained for both LINE-1 mRNA and ORF1p, providing proof-of-principle that EV LINE-1 may serve as a “liquid biopsy” of LINE-1 tissue levels. While EV ORF1p can be readily detected in vitro and in vivo, EV LINE-1 mRNA was more difficult to detect, a limitation likely caused by the stoichiometry of LINE-1 ribonucleoprotein complex formation, in which one 6 kb LINE-1 mRNA molecule binds up to 240 ORF1p molecules and one molecule of ORF2p [29]. In our studies, ORF1p was the target protein of interest given its high abundance compared to ORF2p, which exhibits low cellular abundance due to deficits during cellular processing [28,33,34]. The finding that ORF1p is present in EVs isolated from cells and plasma, and detected as a dimer, was intriguing. We and others have previously reported the presence of an ORF1p dimer in cells, but the factors that dictate the appearance of the dimer relative to the monomer or the trimer have remained elusive [24,26,28]. While ORF1p dimers may represent partially denatured trimers [35], our data argue against this interpretation as dimers were selectively enriched in EVs and segregated from other LINE-1 products. 

Given our interest in using LINE-1 readouts as biomarkers, we first explored EV LINE-1 cargo in the plasma of ostensibly healthy subjects to evaluate inter-individual variability. Our investigation revealed considerable heterogeneity in LINE-1 expression between individuals, with the source of variability likely reflecting genetic, environmental, and lifestyle interactions that influence circulating and tissue LINE-1 levels. For ostensibly healthy subjects, we identified sex-specific and possibly racial/ethnic differences. Females had higher EV LINE-1 levels than males and African Americans displayed wider ranges of EV LINE-1 values compared to Hispanics. These results are consistent with previous studies showing that males exhibit higher levels of LINE-1 hypermethylation than females [36,37] and that LINE-1 methylation can vary as a function of race and ethnicity [38]. Moreover, we recently found greater numbers of ORF1p+ cells in African American lung cancer patients than Caucasians, with suggestive evidence that this difference may be related to lower survival rates in African Americans [36]. Together, our findings confirm previous observations [24] and lend support to the conclusion that LINE-1 expression in plasma and by extension in EVs can be influenced by phenotypic traits [36,37,38]. 

As expected, levels of LINE-1 analytes in subjects with lung cancers were broadly distributed as a function of cancer stage and histological type. We found that mean EV LINE-1 mRNA increased with cancer stage and that higher LINE-1 levels were seen in squamous cell carcinomas. These results are consistent with other studies showing altered LINE-1 DNA methylation in NSCLC leading to variable retroelement expression [7]. Saito et al. (2010) examined tumor LINE-1 methylation status in 379 cases of NSCLC and found that LINE-1 hypomethylation increases as a function of cancer stage and that squamous cell carcinoma, an NSCLC subtype linked to smoking [39], exhibited lower median levels of LINE-1 methylation compared to adenocarcinomas [7]. Consistent with these findings, the H520 squamous cell carcinoma line exhibited the highest levels of LINE-1 expression. In contrast, the lowest LINE-1 levels were seen in adenocarcinoma patients and the adenocarcinoma A549 cell line. Previous studies have shown that LINE-1 may be particularly useful in diagnostics for early stage cancers, as Stages IA and IB exhibit disparate levels of DNA methylation [7] and LINE-1 hypomethylation can be used to identify early NSCLC among current smokers [18]. 

A recent study by Taylor et al. (2023) examined circulating ORF1p levels in plasma using a novel single-molecule array (Simoa) assay and found that plasma ORF1p was elevated in individuals with various cancers, including NSCLC lung cancer [40]. Among these individuals, a greater proportion of squamous cell carcinomas were LINE-1 positive compared to other histological types. This study also confirmed the presence of ORF1p within EVs; although, it also reported an abundance of free-floating ORF1p. While our methodology detected EV ORF1p in healthy subjects, using the Simoa assay, detection greatly depended on the protein sequence of the capture/detection nanobody. As such, differences may be explained by methodology-related differences in the detection of multimeric forms of ORF1p. 

An additional consideration regarding the diagnostic use of LINE-1 EVs in complex matrices such as plasma is the relative enrichment of EVs from target tissues compared to other sources. We hypothesize that the bulk of LINE-1 readily detected in plasma originates from tissues that exhibit constitutive LINE-1 expression such as the brain [41], esophagus, prostate, stomach, or heart muscle [42].

Our findings raise significant questions regarding the functional consequences of circulating EV LINE-1. We and others have noted that EVs containing LINE-1 products possess reverse transcriptase activity [22,23,24] and are capable of modifying the DNA of recipient cells [23]. Enhanced EV LINE-1 export from tumors may therefore increase the potential for genomic aberrations and transformation of near or distant tissues that take up these EVs. Further mechanistic studies will be required to explore these dynamics. Future studies can examine whether the depletion of circulating NSCLC EVs reduces cancer progression. 

In conclusion, our findings suggest that measurements of LINE-1 analytes in EVs may serve as a proxy for tracking changes in NSCLC. Efforts to understand the diagnostic potential of LINE-1 as a cancer biomarker will require the study of large cohorts along with the monitoring of environmental and lifestyle factors that may influence circulating LINE-1 levels.

## 4. Materials and Methods

### 4.1. Tissue Culture

Cell lines were purchased from the American Type Culture Collection (ATCC, Manassas, VA, USA) and confirmed to be free of mycoplasma contamination [43]. All cell lines were cultured at 37 °C, 5% CO_2_ in a humidified environment, and seeded to 70% confluence for 24 h. Cell lines were cultured as follows: BEAS-2B cells were cultured in LHC-9 medium, A549 cells in DMEM plus 10% fetal bovine serum (FBS), and the remaining lines in RPMI plus 10% FBS (Gibco/ThermoFisher Scientific, Waltham, MA, USA). To prepare for EV collection, cells were washed with DPBS and incubated in media containing EV-depleted FBS (Gibco/ThermoFisher Scientific) for 48 h. Conditioned media were collected, frozen, and processed for EV isolation upon thawing. To study inducible LINE-1 responses, 2 × 10^6^ H460 cells were seeded in 150 mm dishes and allowed to attach overnight. The next day, cells were washed with DPBS and incubated with RPMI medium containing 10% EV-depleted FBS plus various concentrations of benzo(a)pyrene (BaP), a lung carcinogen and known inducer of LINE-1 in normal and transformed lung epithelial cell lines [24,44,45], dissolved in 0.2% DMSO. Conditioned media were collected after 48 h and processed for EV isolation and measurement of cellular viability by trypan blue dye exclusion. Unconditioned media (UCM) was used as media control. Conditioned EV-free media (EFM) were used to measure the relative abundance of free versus EV-associated LINE-1 cargo.

### 4.2. EV Isolation and Characterization

For in vitro experiments, approximately 120 mL of conditioned media was centrifuged for 5 min at 500× *g* and concentrated to <12 mL using Centricon Plus-70 100 kDa MWCO centrifugal filter units (Millipore-Sigma, Darmstadt, Germany). The concentrate was centrifuged at 21,800× *g* for 30 min to pellet cellular debris, apoptotic bodies, and large vesicles. EVs were isolated using Polyethylene Glycol (PEG) or ultracentrifugation, as indicated. For PEG isolation, PEG-MW 6000 (Millipore-Sigma) was added to the supernatant to 10% *w*/*v*, incubated on ice for 15 min, and centrifuged at 21,800× *g* for 15 min. To remove protein contaminants, the EV pellet was resuspended in PBS, transferred to a clean tube, and reprecipitated in 10% PEG. PEG EV pellets were resuspended in PBS and confirmed to be free of debris using Nanosight Nanoparticle Tracking Analysis (NTA) (Malvern Panalytical, Malvern, UK). For ultracentrifugation, concentrated media were centrifuged at 100,000× *g* for 2 h at 4 °C. Pellets were washed in cold PBS and the centrifugation was repeated before resuspension of EV pellets in PBS. Protein was used to normalize EV input in in vitro experiments. Nanosight NTA of cell media preparations was performed by the Center for Nanotechnology in Drug Delivery at the University of North Carolina Chapel Hill. EVs in PBS were diluted and quantified with a NanoSight NS500 (Malvern) equipped with red laser illumination (532 nm). Each sample was read five times, using four samples per treatment. 

### 4.3. Collection of Plasma EVs

Plasma was purchased from BioIVT (Westbury, NY, USA) from ostensibly healthy donors who were screened verbally to rule out serious health conditions such as prior metastatic disease, diabetes, or other inflammatory diseases. Plasma was collected using sodium citrate. Plasma was also obtained from patients undergoing surgical resection for a lung lesion at Houston Methodist Hospital. The collection of plasma and nodular tissue was approved by the Institutional Review Board at Houston Methodist Research Institute (Protocol # 00004763). Heparinized venous blood was collected prior to nodule resection and the final pathology report was obtained to identify tumor characteristics and staging for each patient. For plasma EV isolation, samples were thawed, and platelets were removed after two centrifugation cycles at 3600× *g*. Cleared plasma was diluted 4× with DPBS and centrifuged at 21,800× *g* for 30 min to pellet debris. The supernatant was ultracentrifuged as described above. EV inputs were normalized by input plasma volume.

### 4.4. Western Blotting

Cells and EVs were lysed in RIPA buffer and subjected to Western blotting as described [46]. Immunoblots were imaged using a Konica Minolta SRX101A film developer (Konica Minolta Medical & Graphic, Inc., Tokyo, Japan). The ORF1p polyclonal antibody used was custom-made using the 14-amino acid N-terminus of the ORF1p protein (ORF1p1-14). The specificity of this antibody has been confirmed in previous studies [47]. The monoclonal ORF1p antibody used was purchased from Millipore-Sigma (MABC1152). Antibodies to ALIX (CST, 92880), CD9 (CST, 13403), Flotillin-1 (CST, 18634), GAPDH (CST, 2118), and Calnexin (MA5-31501) were obtained commercially via Cell Signaling Technology (Danvers, MA, USA) and ThermoFisher.

### 4.5. ELISA

ORF1p was measured using a competitive indirect ELISA, where indicated, as this platform is not subject to PEG interference (Appendix A) [27]. Briefly, pre-blocked streptavidin-coated plates (Pierce/ThermoFisher) were incubated with biotinylated ORF1p1-14. After washing with PBS/0.01% Tween, diluted plasma, standards, and primary antibody were mixed, added to the plates, and incubated for 1 h. Standards were generated using known quantities of ORF1p1-14 and a matrix control of EV pellets derived from murine cells. After washing and incubation with HRP-linked secondary antibody (ThermoFisher), the ELISA was developed using TMB (Pierce/ThermoFisher) substrate and subsequently quenched with sulfuric acid. Absorbance was read at 450 nM. 

### 4.6. LINE-1 mRNA Cargo and Quantification

EVs from H520 cells were used to evaluate LINE-1 mRNA content. Briefly, RNAseA was added to EVs in DPBS and incubated for 10 min at room temperature. RNA Secure (Invitrogen/ThermoFisher) was used for RNA extraction. EV protein was used to normalize EV input across cell lines and treatments. RNA was extracted from EV pellets using a Quick RNA kit (Zymo, Irvine, CA, USA) with DNAseI digestion (Turbo DNA Free, Invitrogen/ThermoFisher). RNA was eluted in equal volumes of nuclease-free water, and 3 μL was used for each RT-qPCR reaction. LINE-1 and β-Actin were quantified using Luna Universal One-Step RT-qPCR (NEB, Ipswich, MA, USA) according to the manufacturer’s protocol, with duplicate reactions for each sample. LINE-1 contains no introns; thus, LINE-1 primers cannot distinguish between residual genomic (g) DNA and cDNA. Thus, each LINE-1 sample was normalized to a matched control lacking reverse transcriptase (RTC) to control for residual gDNA. As exon-junction spanning β-Actin primers do not amplify gDNA, the β-Actin signal was normalized to a non-template control (NTC). All primers had amplification efficiencies of >90%. LINE-1 primer: Forward 5′ACACCTATTCCAAAATTGACCAC 3, Reverse 5′ TTCCCTCTACACACTGCTTTGA 3′, and probe 5′ TG GAAACTGAACAACCTGCTCCTGA 3′. β-Actin primers: Forward 5′ CTGGCACCCAGCACAATG 3′, Reverse 5′ GCCGATCCACACGGAGTACT 3′, Probe: 5′ ATCAAGATCATTGCTCCTCCTGAGCGC 3′.

### 4.7. Immunohistochemistry

Sections of human lung tumors and patient-matched non-tumor adjacent tissues were deparaffinized in xylene and rehydrated in a graded ethanol series. HistoZyme was performed for antibody retrieval. Endogenous peroxidase was inactivated by treating tissue sections with 3% H_2_O_2_/methanol for 15 min. The slides were washed five times for five min each and the sections blocked with 5% goat serum containing 0.1% Triton X-100 at room temperature for 1 h, followed by labeling with primary ORF1 antibody (Millipore) at 1:200 at 4 °C overnight. The primary antibody was labeled with HRP-conjugated secondary antibodies at 1:200 for 1 h at room temperature. The signal was visualized using a DAB substrate kit. Stained sections were dehydrated and mounted. Images were taken with Nikon eclipse Zyla SCMOS microscope at 40× magnification.

### 4.8. Statistical Analysis

Statistical analyses were performed using GraphPad Prism 8.1.2. A *p*-value of less than 0.05 was considered significant. EV LINE-1 mRNA enrichment was compared to background levels using a one-tail *t*-test. The association between cellular and EV ORF1p with or without BaP treatment was established using simple linear regression. The relationship between cellular and EV mRNA was tested using Spearman’s non-linear regression. BaP treatments were compared to the DMSO control using a one-way ANOVA with Dunnett’s multiple comparisons or Kruskal–Wallace test with Dunn’s multiple comparisons, as indicated. The relationship between EV LINE-1 mRNA and ORF1p content in healthy plasma EVs was assessed by simple linear regression. The mean EV LINE-1 mRNA and ORF1p content were compared between males and females and African American and Hispanic American subjects using unpaired *t*-tests. Whole plasma ORF1p was compared to EV ORF1p levels using a simple linear regression.

## Figures and Tables

**Figure 1 ijms-25-01169-f001:**
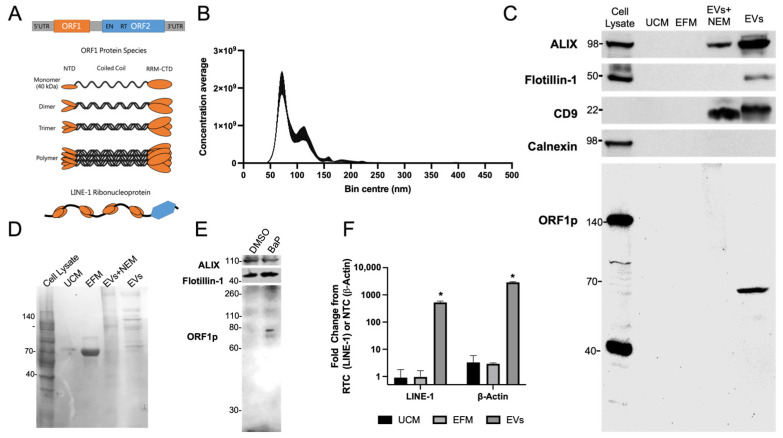
EV isolation and identification of LINE-1 mRNA and ORF1p EV cargo. (**A**) LINE-1 element structure, proteins, and RNP. Full length LINE-1 is approximately 6 kb in length and consists of 5′ and 3′ untranslated regions (UTRs) and two proteins, ORF1p (orange) and ORF2p (blue). ORF1 is a nucleic acid binding protein consisting of an alpha helix in between the N- and C- terminal domains (NTC, CTD), with an RNA recognition motif (RRM) near the CTD. Coiled-coil interactions facilitate the formation of higher order multimers and polymers of ORF1p. ORF2p is approximately 150 kDa and contains both endonuclease (EN) and reverse transcriptase (RT) domains. Both ORF1p and ORF2p exhibit strong affinity for the LINE-1 mRNA, and together form ribonucleoproteins (RNPs) containing both the template and enzymatic machinery required for target-site primed reverse transcription. EVs from H520 cells were collected using ultracentrifugation along with unconditioned media (UCM) and EV-free media (EFM). Equal quantities of EV protein, cell lysate protein, and an equivalent volume of UCM and EFM (30 µL) were examined using SDS-PAGE and Western blotting. (**B**) Nanoparticle tracking analysis was performed to examine the size and abundance of EV isolates. Mean and SE of five measurements shown. (**C**) The EV markers, ALIX, Flotillin-1, and CD9 were examined in the preparations. The absence of calnexin, an endoplasmic reticulum protein, indicates absence of contaminating proteins. The presence of LINE-1 protein, ORF1p, was examined using an in-house polyclonal antibody. ORF1p forms multimers resistant to denaturation during reducing SDS-PAGE. Unlike cell lysates, an ORF1p dimer is predominant in EVs. EV protein treated with the irreversible reducing agent N-ethylmaleimide (NEM) was included for reference. (**D**) Ponceau staining was completed to confirm protein loading. (**E**) H460 cell cultures were treated with a LINE-1 inducer, benzo[a]pyrene (BaP) and the EVs were collected and examined via SDS-PAGE, which resulted in the appearance of the ORF1p dimer. (**F**) To demonstrate the presence of LINE-1 mRNA within EVs, intact EVs were treated with RNAseA prior to lysis and RNA isolation. Similar treatments were performed with UCM and EFM controls for comparison. β-Actin is shown as a positive control. LINE-1 was normalized to RTC and β-Actin to NTC. Mean, SD shown. *N* = 2. * FC > 1, above background; *p* < 0.05. One-tail *t*-test.

**Figure 2 ijms-25-01169-f002:**
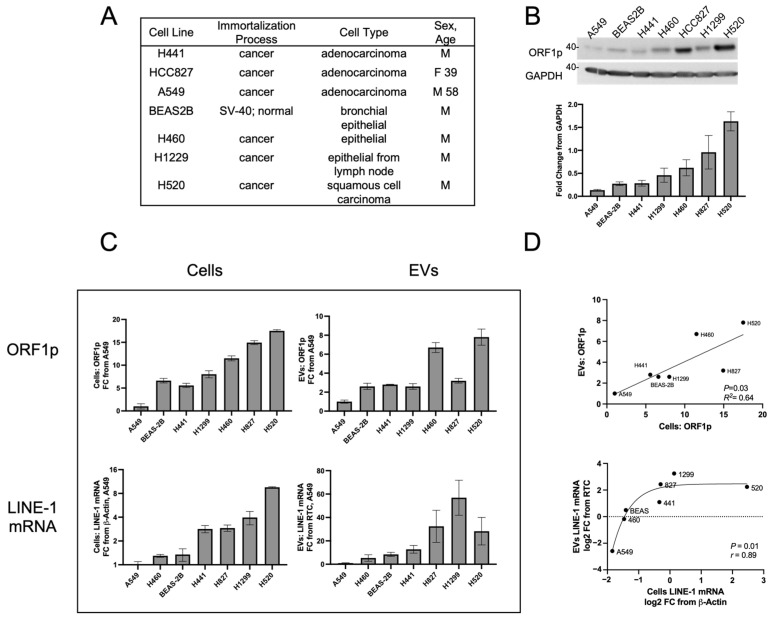
LINE-1 levels in cells and Evs across a panel of NSCLC cell lines. (**A**) Characteristics of the lung epithelial cell lines used in this study. (**B**) Relative expression of ORF1p as assessed by Western blotting (top) and densitometry (bottom). To improve the sensitivity of EV detection and overcome PEG interference with SDS-PAGE (Appendix A), an in-house ORF1p ELISA was used. EV abundance was normalized by protein levels. All measures are presented as fold change relative to A549 cells or Evs, which exhibited the lowest LINE-1 levels. (**C**) ORF1p was measured by ELISA in cells ((**C**), top left) and Evs ((**C**), top right). LINE-1 mRNA was quantified using One-Step qPCR within cells ((**C**), bottom left) and Evs ((**C**), bottom right). (**D**) The relationship between cellular and EV ORF1p (top) was assessed using Pearson’s linear regression (*p* = 0.03, *R*^2^ = 0.64). The relationship between cellular and EV mRNA levels ((**D**), bottom) was assessed using Spearman’s non-linear regression analysis (*r* = 0.89, *p* = 0.01). Means ± SE shown, *N* = 3.

**Figure 3 ijms-25-01169-f003:**
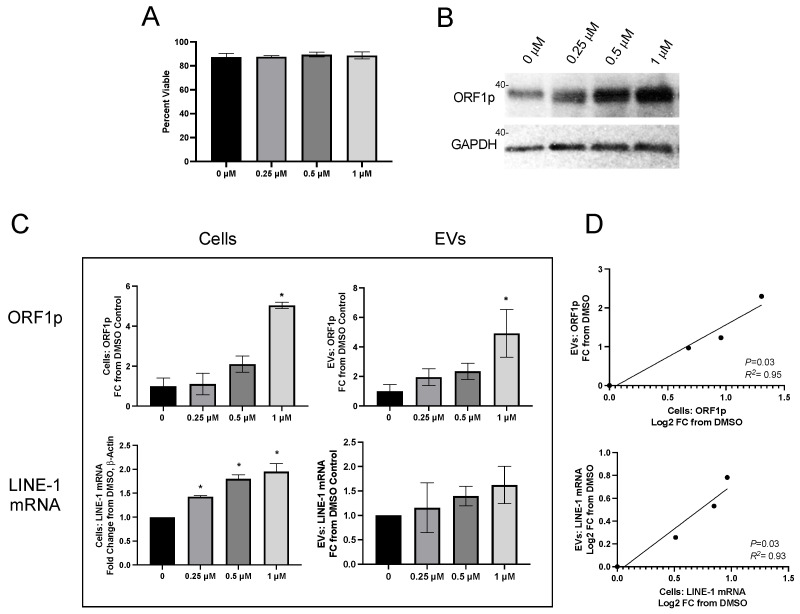
LINE-1 induction by BaP increases LINE-1 mRNA and ORF1p export in EVs. H460 cells were challenged with different concentrations of the LINE-1 inducer, benzo[a]pyrene (BaP), for 48 h before cells and conditioned media were collected. (**A**) Cell viability was unaffected by BaP treatment, as measured by Trypan Blue exclusion. (**B**) Representative Western blot depicting cellular ORF1p expression across BaP treatments. (**C**) ORF1p was measured by ELISA in cells ((**C**), top left) and EVs ((**C**), top right). LINE-1 mRNA was quantified using One-Step qPCR within cells ((**C**), bottom left) and EVs ((**C**), bottom right). BaP treatments were compared to DMSO control by one-way ANOVA with Dunnett’s multiple comparisons ((**A**,**C**), top row, bottom left) and Kruskal–Wallace with Dunn’s multiple comparisons ((**C**), bottom right) (* *p* < 0.05). (**D**) LINE-1 values were compared between cells and EVs using linear regression. Means ± SEM are shown. *N* = 4.

**Figure 4 ijms-25-01169-f004:**
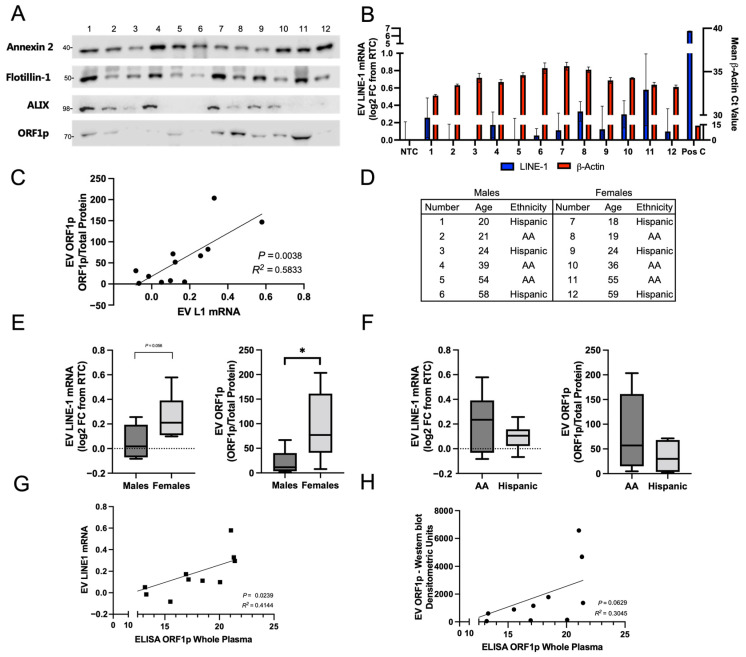
LINE-1 Analyte Profiles in Human Plasma-Derived EVs from Ostensibly Healthy Subjects. Plasma from 12 ostensibly healthy human subjects was used in these experiments. EVs were isolated from plasma by ultracentrifugation and resuspended in equal volumes of PBS. EV inputs were normalized by the total volume of input plasma (3 mL). Half of the preparation was used for protein analysis via Western blotting and the other for LINE-1 mRNA quantification. (**A**) Immunoblot of plasma EVs. Annexin 2, Flotillin-1, and ALIX were used as positive controls, as they are enriched in many types of EVs. An ORF1p dimer was detected in EVs. (**B**) EV mRNA quantification. LINE-1 (left y-axis) was expressed as an FC from RTC. β-Actin detection was used as a positive control (right y-axis). (**C**) The relationship between EV LINE-1 mRNA and ORF1p content was assessed by simple linear regression. (**D**) Matched age (within 2 years), sex, and race of plasma donors. African American, “AA”. Mean EV LINE-1 mRNA ORF1p content were compared between males and females (**E**) and black and Hispanic subjects (**F**) using unpaired *t*-tests. The patterns in (**C**,**E**,**F**) exhibited no association with slight variations in EV protein levels (Appendix A). * *p* < 0.05. F. Whole plasma ORF1p measurements were examined as a predictor of EV LINE-1 content. ORF1p was measured by ELISA in 120 μL of whole plasma and these measures were then compared to measures of EV LINE-1 mRNA (**A**) and ORF1p (**B**) using a simple linear regression. Bulk plasma ORF1p in ostensibly healthy subjects using ELISA was compared to LINE-1 mRNA (**G**) and EV ORF1p (**H**).

**Figure 5 ijms-25-01169-f005:**
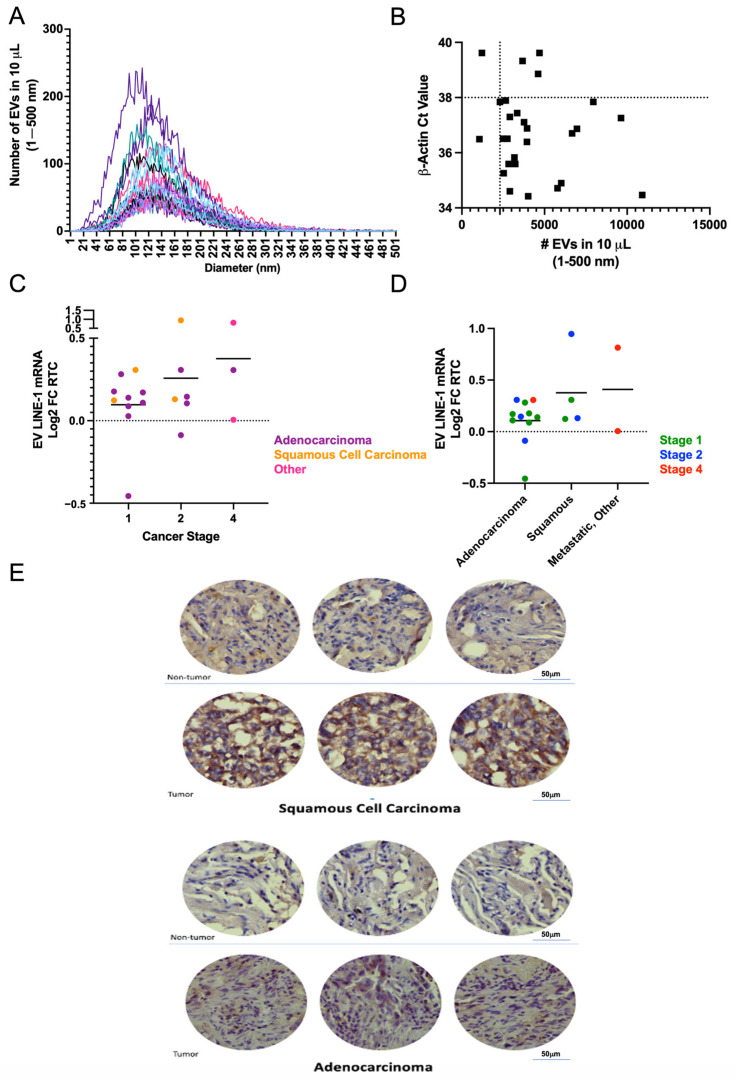
EV LINE-1 mRNA Levels in NSCLC Patients Stratified by Clinical Stage and Histopathology. Plasma was collected from patients with lung nodules who were undergoing surgical resection for suspected lung cancer. Platelets were removed from 250 μL plasma and EVs collected via ultracentrifugation and mRNA quantified by One-Step RT-qPCR. Three patients were found to have granulomatous disease and not lung cancer and were excluded from the analyses. (**A**) To ensure adequate sensitivity, the EV composition of samples was verified by nanoparticle tracking analysis. Different colors depict different preparations. (**B**) Samples were excluded from further analysis if the β-Actin Ct was greater than 38 (horizontal line) or if they possessed fewer than 2300 total particles between 1 and 500 nm (vertical line). (**C**,**D**) EV LINE-1 mRNA levels were stratified based on cancer stage or histopathology, respectively. Within each strata subjects were color coded as denoted. Column means are indicated by the black horizontal lines. (**E**) Sections of human lung tumors and patient-matched non-tumor adjacent tissues were deparaffinized in xylene and rehydrated in a graded ethanol series. HistoZyme was performed for antibody retrieval. ORF1 antibody (Millipore) at 1:200 signal was visualized using a DAB substrate kit. Images were taken with Nikon eclipse Zyla SCMOS microscope at 20, 40, and 60× magnifications. Immunohistochemical analysis of squamous cell carcinomas (SCC) and adenocarcinomas (LUAD) compared to adjacent non-tumor tissues revealed that LINE-1 ORF1p is highly expressed in lung cancer tissues and that protein expression may be higher in SCC than in LUAD. The images shown are representative of 38 sample sets that included 19 tumors (9 SCC and 10 LUAD) and 19 patient-matched non-tumor adjacent tissues. The SCC images are for a Black, 74-year-old female, former smoker, with non-keratinizing tumor at pathology Stage 2B. The LUAD images are for a Black, 55-year-old female, smoker at the time of surgery, with acinar, moderately differentiated tumor at pathology Stage 1A3.

## Data Availability

The authors confirm that the data supporting the findings of this study are available within the article and Appendix A.

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
