# Peer review of "Long Interspersed Nuclear Element-1 Analytes in Extracellular Vesicles as Tools for Molecular Diagnostics of Non-Small Cell Lung Cancer"

_ijms, 2024, doi:10.3390/ijms25021169_

Round 1
Reviewer 1 Report
Comments and Suggestions for Authors
In this work, Bowers et al. studies LINE-1 analytes (mRNA and ORF1p protein) in extracellular vesicles (EVs) in cancer types, including non-small cell lung cancers (NSCLCs). They found that whole plasma ORF1p levels correlated with EV ORF1p levels, indicating that most circulating LINE 1 proteins are contained within EVs. LINE 1 mRNA levels were elevated in patients with advanced cancer stages and in patients with squamous cell carcinoma and metastatic tumors compared to adenocarcinomas. These data support LINE-1 analytes as a biomarker for NSCLC.
There are a few concerns.
First, in Figure 1C, ORF1p proteins were in monomers and tetramers in H520 cell lysates but in dimers in EVs. Any reasons for this discrepancy? Are they common in other lung cancer cell lines listed in Figure 2A?
Second, MW markers shall be shown in Figures 2B and 3B, so the ORF1p isoforms can be seen.
Third, Taylor et al. published a paper “Ultrasensitive Detection of Circulating LINE-1 ORF1p as a Specific Multicancer Biomarker” in Cancer Discovery (2023). In the Taylor paper, ORF1p was detected in lung cancer patients’ plasma. Please include this paper in the Discussion.
Last, this work did not show much data on lung cancer progression. Please revise “LINE-1 analytes in plasma EVs may serve to monitor the activity of LINE-1 retroelements during malignant progression of lung cancer” in the Abstract.
Author Response
|
1. Summary |
|
|
|
Thank you very much for taking the time to review this manuscript. Please find the detailed responses below and the corresponding revisions/corrections highlighted/in track changes in the re-submitted files.
|
||
|
2. Point-by-point response to Comments and Suggestions for Authors |
||
|
Comments 1: In this work, Bowers et al. studies LINE-1 analytes (mRNA and ORF1p protein) in extracellular vesicles (EVs) in cancer types, including non-small cell lung cancers (NSCLCs). They found that whole plasma ORF1p levels correlated with EV ORF1p levels, indicating that most circulating LINE 1 proteins are contained within EVs. LINE 1 mRNA levels were elevated in patients with advanced cancer stages and in patients with squamous cell carcinoma and metastatic tumors compared to adenocarcinomas. These data support LINE-1 analytes as a biomarker for NSCLC. There are a few concerns. First, in Figure 1C, ORF1p proteins were in monomers and tetramers in H520 cell lysates but in dimers in EVs. Any reasons for this discrepancy? Are they common in other lung cancer cell lines listed in Figure 2A? |
||
|
Response 1: The observed differences in ORF1p complex assembly between cell lysates and EVs of ORF1p have been observed in other cancer cell lines, including NCI H460 cells. In fact, Figure 2B in a previous publication from the laboratory (Bowers et al., 2022) a full-length Western blot comparing ORF1p profiles in cell lysates versus EVs we showed that H460 cell lysates predominantly showed monomers, while EVs exhibited the dimeric complex. In that same manuscript, Figure 1E showed that the ORF1p dimer was inducible in H460 cells by treatment with the lung carcinogen benzo(a)pyrene. Among all cell lines examined to date in previous studies, H520s and H460s show the highest levels of ORF1p in EVs. ORF1p expression and EV ORF1p content in other cell lines tested was reduced and required more sensitive ELISA measurements for accurate detection. Our previous studies also showed that the ORF1p dimer is readily detected in human plasma. To address the question of the reviewer, we have added the following statements in the discussion: “While ORF1p dimers may represent partially denatured trimers [40], our data argue against this interpretation as dimers were selectively enriched in EVs and segregated from other LINE-1 products.”
|
||
|
Comments 2: Second, MW markers shall be shown in Figures 2B and 3B, so the ORF1p isoforms can be seen. Response 2: The molecular weight markers were added to the strip blots as requested.
|
||
|
Comments 3: Third, Taylor et al. published a paper “Ultrasensitive Detection of Circulating LINE-1 ORF1p as a Specific Multicancer Biomarker” in Cancer Discovery (2023). In the Taylor paper, ORF1p was detected in lung cancer patients’ plasma. Please include this paper in the Discussion. Response 3: A paragraph discussing this paper has been added to the discussion.
|
||
|
Comments 4: Last, this work did not show much data on lung cancer progression. Please revise “LINE-1 analytes in plasma EVs may serve to monitor the activity of LINE-1 retroelements during malignant progression of lung cancer” in the Abstract. Response 4: The word “progression” has been removed and the sentence now reads “…monitor the activity of LINE-1 retroelements in lung cancer.”
|
||
Please see the revised manuscript in the attachment.

Reviewer 2 Report
Comments and Suggestions for Authors
The author detected the expression of LINE-1 in EVs and analyzed their correlation in NSCLC. This study gives insight into understanding role of EVs in tumors. I have some concerns:
1. The organization of some figures and results should be checked, for example, Page 4. Line 204 should be Figure 1, and should be given a sub-title. And the Figure 1a-1f should be introduced sequentially.
2. Why the authors choose NSCLCs to study should be explained.
3. Are EV LINE-1 RNA correlates with as genomic DNA methylation and oncogene expression in cell lines or patient cells?
4. EV profile and their correlation with cancer progression should be analyzed in mouse model, and it is interesting to see whether EV-LINE1 level increase with disease progression in the same cancer patients.
5. What’ s the rational for analyzing LINE-1 concentration in EV of health donors ?
6. Could depletion of cancer cell-derived EVs reduce tumour progression?
7. Figure 5 E missed scale bars.
8. The discussion section is very long and redundant and should be refined, results and backgrounds should not be repeated, and the meaning of this study should be further discussed.
9. The fonts and size in the figure should be same throughout the manuscript.
Author Response
|
1. Summary |
|
|
|
Thank you very much for taking the time to review this manuscript. Please find the detailed responses below and the corresponding revisions/corrections highlighted/in track changes in the re-submitted files. |
||
|
2. Point-by-point response to Comments and Suggestions for Authors |
||
|
The author detected the expression of LINE-1 in EVs and analyzed their correlation in NSCLC. This study gives insight into understanding role of EVs in tumors. I have some concerns: Comments 1: The organization of some figures and results should be checked, for example, Page 4. Line 204 should be Figure 1, and should be given a sub-title. And the Figure 1a-1f should be introduced sequentially. |
||
|
Response 1: We appreciate the feedback and have made proper corrections. We now cite Figure 1A in the introduction. |
||
|
Comments 2: Why the authors choose NSCLCs to study should be explained. Response 2: LINE-1 dysregulation has been shown to be especially predominant in lung cancers. Moreover, additional molecular biomarkers are desperately needed for NSCLC to combat high false positive screening results. To make this clearer, we added the sentence in the introduction, “Non-small cell lung cancers (NSCLCs) are strongly impacted by LINE-1 dysregulation.” This immediately precedes the further explanation that “LINE-1 reactivation is prevalent during early stages, non-small cell lung cancers (NSCLCs), especially in smokers6,7,18, and the genome of NSCLC is strongly affected by LINE-1 insertions13,19.” |
||
|
Comments 3: Are EV LINE-1 RNA correlates with as genomic DNA methylation and oncogene expression in cell lines or patient cells? Response 3: While previous studies in our laboratory have measured oncogenes in several individual cell lines, we did not take these measures across the entire cell panel in the present study. This is an excellent idea for future studies. |
||
|
Comments 4: EV profile and their correlation with cancer progression should be analyzed in mouse model, and it is interesting to see whether EV-LINE1 level increase with disease progression in the same cancer patients. Response 4: We agree that this would be an interesting comparison. One challenge with EV work in mouse models is that it can be difficult to obtain enough plasma in adequate quantities for analysis and the fact that LINE-1 in mice is regulated differently from humans. |
||
Comments 5: What’s the rationale for analyzing LINE-1 concentration in EV of health donors?
Response 5: We examined EV LINE-1 in healthy individuals to have a sense of the degree of inter-individual variability between subjects. To clarify this point, we added several sentences to the results section:
“We first examined the abundance of LINE-1 ORF1p in ostensibly healthy subjects to evaluate the degree of inter-individual variability.”
“Together these results reveal that EV LINE-1 content varies considerably between individuals. Importantly, these findings can inform the creation of larger range finding studies to evaluate EV LINE-1 patterns in cancer patients.”
Comments 6: Could depletion of cancer cell-derived EVs reduce tumour progression?
Response 6: Given that numerous studies have shown that cancer cell EVs play a role in tumor progression and metastasis, this is a very exciting possibility! We have added a statement to this effect in the discussion section.
Comments 7: Figure 5 E missed scale bars.
Response 7: This has been corrected in the revised manuscript.
Comments 8: The discussion section is very long and redundant and should be refined, results and backgrounds should not be repeated, and the meaning of this study should be further discussed.
Response 8: The discussion has been revised as suggested.
Comments 9: The fonts and size in the figure should be same throughout the manuscript.
Response 9: This has been corrected as suggested.
Please see the revised manuscript in the attachment.
